# Post-Marketing Active Surveillance of Adverse Reactions Following Influenza Cell-Based Quadrivalent Vaccine: An Italian Prospective Observational Study

**DOI:** 10.3390/vaccines9050456

**Published:** 2021-05-04

**Authors:** Pasquale Stefanizzi, Sara De Nitto, Giuseppe Spinelli, Sabrina Lattanzio, Paolo Stella, Domenica Ancona, Maria Dell’Aera, Margherita Padovano, Savino Soldano, Silvio Tafuri, Francesco Paolo Bianchi

**Affiliations:** 1Department of Biomedical Science and Human Oncology, Aldo Moro University of Bari, 70124 Bari, Italy; pasquale.stefanizzi@uniba.it (P.S.); sara.denitto@asl.bari.it (S.D.N.); giuseppe.spinelli@uniba.it (G.S.); sabrina.lattanzio@uniba.it (S.L.); dr.francesco.bianchi@gmail.com (F.P.B.); 2Regional Center for Pharmacovigilance, 52100 Puglia, Italy; p.stella@regione.puglia.it (P.S.); domenica.ancona@aslbat.it (D.A.); 3Department of Pharmacy, Bari Policlinico General Hospital, 70124 Bari, Italy; Maria.dellaera@policlinico.ba.it (M.D.); margherita.padovano@policlinico.ba.it (M.P.); 4Health Hospital Management, Bari Policlinico General Hospital, 70124 Bari, Italy; savino.soldano@policlinico.ba.it

**Keywords:** post-marketing active surveillance, vaccination, influenza, causality assessment, adverse reactions

## Abstract

Since the influenza season 2018/19, the Italian Ministry of Health recommended a dose of cell-based quadrivalent vaccine (Flucelvax Tetra) for HCWs (healthcare workers), because this vaccine seemed more efficacious in the prevention of AH3N2 virus. Due to the lack of pre-registration data, the safety profile of this new vaccine must be investigated in post-marketing surveillance. The aim of our study is to evaluate, through a post-marketing active surveillance program developed during the 2019/20 influenza season, any Adverse Events Following Immunization (AEFIs) that happened in the 7 days after immunization with Flucelvax Tetra. The study was carried out in a sample of HCWs of Policlinico General University-Hospital (Apulia, South Italy). AEFIs were classified as ‘serious’ or ‘not serious’ according to the WHO (World Health Organization) guidelines; the WHO causality assessment algorithm was applied to classify serious AEFIs. A total of 741 HCWs were enrolled, and 430 AEFIs (reporting rate: 58.0 (95%CI: 54.4–61.6) × 100 enrolled) were recorded. Of these, 429 of 430 (99.8%; reporting rate: 57.8 (95%CI: 54.2–61.5) × 100 enrolled) were classified as not serious and one (0.2%; reporting rate: 0.13 (0.03–0.75) × 100 enrolled) was classified as serious. Local reactions were the adverse reaction reported most frequently (88%); regarding the serious AEFI, causality assessment excluded the causal link with the administration of the vaccine. All the AEFIs resolved without sequelae. Flucelvax Tetra showed a profile of high safety. Due to their characteristics of greater sensitivity than passive surveillance, active surveillance programs can be useful in defining the safety profiles of a given vaccine/drug in certain population subgroups.

## 1. Introduction

The vaccination of healthcare workers (HCWs) is an effective measure of individual and collective disease protection. It protects both HCWs from the occupational risk of infectious disease and patients from the risk of infection in the nosocomial environment [1].

Since HCWs care for people at high risk of influenza-related complications (such as patients affected by chronic disease, cancer, or immunodeficiency), it is especially important that they are vaccinated annually. Indeed, high vaccine coverage among HCWs has been shown to reduce the rates of influenza morbidity and mortality among their patients [2].

In Italy, the vaccination of HCWs is provided by Legislative Decree 9 April 2008 n. 81 [3]. Official recommendations for the immunization of HCWs are also part of the National Immunization Plan and the annual influenza prevention guidelines provided by the Italian Ministry of Health. In both, HCWs are among the risk categories for which influenza vaccination is strongly recommended [4,5]. Italy also recommends the active offer of influenza vaccine to HCWs every year during the influenza season (from October to December), with the vaccination strategy managed by the hospital director and occupational physician. Since the influenza season 2018/19, the Italian Ministry of Health recommended a dose of cell-based quadrivalent vaccine (Flucelvax Tetra, Seqirus S.r.l., Siena, Italy) [6] for HCWs. In October 2018, the vaccine was authorized for use from 9 years of age (single dose of 0.5 mL) by the European Medicines Agency (EMA), and in December of the same year, it obtained the final approval by the European Commission [7,8].

This typology of vaccine has shown greater protection against flu or flu-like illness compared to standard-dose egg-based vaccines [6]. The safety of Flucelvax Tetra in adults aged 18 years and older was evaluated in a randomized, controlled study in which 1334 subjects received it. In this clinical study, similar rates of local and systemic adverse reactions were reported in subjects who received Flucelvax Tetra and those who received comparator, which is a trivalent influenza vaccine produced in mammalian cells. The most commonly reported reactions (≥10%) in subjects receiving Flucelvax Tetra were pain at the injection site (34%), headache (14%), fatigue (14%), myalgia (14%), erythema (13%), and induration at the injection site (10%). The incidence of some adverse reactions was considerably lower in subjects aged ≥65 years compared to subjects aged between 18 and 65 years [9].

In the post-marketing life of the vaccines, the World Health Organization (WHO) and National and International Drug Authorities (such as, for Italy, the Italian Drug Authority (AIFA)) monitor their safety by collecting and analyzing reports of adverse events (passive surveillance) or by performing specific active surveillance programs [10]. Passive surveillance involves consumers (immunized people or their parents) and/or healthcare professionals; they recognize and spontaneously subdue AEFIs to health authorities. This step is followed by investigation and response when required [11]. Although international guidelines recommended that all AEFIs must be detected, passive post-marketing surveillance is affected by under-reporting (especially for non-serious adverse events), biased reporting (with several difficulties to distinguish coincidental from causal events), delayed notifications, and incomplete reporting information [12].

The aim of our study is to evaluate through a post-marketing active surveillance program, any Adverse Events Following Immunization (AEFIs) in the 7 days after immunization with Flucelvax Tetra in a sample of HCWs of Policlinico General University-Hospital (Apulia, Italy). In fact, once it has been put on the market, it is essential to constantly monitor and update the safety and efficacy profile of the vaccine product through specific post-marketing surveillance programs. Especially in the case of recently introduced vaccines and/or produced with innovative techniques, it becomes essential to start ad hoc active surveillance programs, which can confirm, strengthen, and/or modify the safety and effectiveness profile of the vaccine product being monitored [13].

## 2. Material and Methods

This was a prospective observational study.

The Bari Policlinico General University-Hospital is the largest hospital in southern Italy. It consists of 50 OUs and 1000 beds and a healthcare staff of almost 4000 people.

During the 2019/20 season, the hygiene department, in collaboration with the Occupational Medicine department, set up, from October to December 2019, an ad hoc clinic and an on-site strategy [14] targeting most of the Policlinico operative units. The clinic was open for about 10 h a day, from Monday to Friday, and it could be visited without an appointment. Furthermore, shortly before the vaccination campaign, specific posters were placed in the Operative Units (OUs) to announce the vaccination schedule. The activities were staffed by Public Health physicians, experts in the field of vaccinology, and residents from the Graduate School of Public Health.

Informed consent was also obtained at the time of vaccination. During the 2019/20 influenza season, vaccinated HCWs received a dose of cell-based quadrivalent vaccine (Flucelvax Tetra) [4], which was administered intramuscularly in the deltoid. The vaccinated individuals were followed up for 7 days in order to detect any adverse effects; a post-vaccination diary was implemented, and vaccinated HCWs were instructed to notify, day by day, any adverse reaction. After 7 days of vaccination, the HCWs were contacted by telephone by the staff of the hygiene department to obtain the information entered in the post-vaccination diary. Adverse reactions reported by HCWs were documented by the Pharmacovigilance Service of the Policlinico Bari General Hospital and entered into the database of the hygiene department. This active surveillance program was not mandatory, and so HCWs could join it on voluntary basis.

The study was notified and approved by the Italian Drug Authority (AIFA).

As provided by Italian law, all adverse events recognized in this study were notified to the AIFA and put in the database of the Italian Pharmacovigilance Network (RNF).

In case of access to the structures of the National Health Service, the health documentation was acquired (hospital discharge card, laboratory, and/or instrumental test reports, specialist consultations, etc.). The data were stored according to privacy law.

WHO guidelines have been used to classify AEFIs as ‘serious’ or ‘not serious’ [15]. They are considered serious in case of death, life threat, in-patient hospitalization, or prolongation of existing hospitalization, persistent or significant disability/incapacity, congenital anomaly/birth defect, or intervention to prevent permanent impairment or damage. Furthermore, in 2016, AIFA published a list of health conditions that must be considered as serious AEFIs [16].

For serious AEFIs, the WHO causality assessment algorithm was applied to classify AEFI as ‘consistent causal association’, ‘inconsistent causal association’, ‘indeterminate’, or ‘not-classifiable’. The causality assessment was carried out separately by two public health physicians who were experts in vaccinology; in case of disagreement, a third physician was consulted [17].

Our study included all the HCWs of Bari Policlinico vaccinated against influenza in the season 2019/20 and that joined the active surveillance program. Personal data, information reported in the post-vaccination diary, health documentation, and the results of the causality assessment were put in a database created with an Excel spreadsheet, and the data were analyzed anonymously using STATA MP16 software. Continuous variables were expressed as the mean ± standard deviation and range, and categorical variables as proportions. A Chi-square test was used to compare categorical variables between groups.

The adverse events reported were grouped into the following categories:Local reactions (pain, redness, swelling, induration at the injection site)Allergic reaction (anaphylaxis, allergic/urticarial reaction)Gastrointestinal symptoms (nausea, vomiting, diarrhea)General malaise (asthenia, myalgia, malaise, drowsiness/insomniaNeurological symptoms (irritability, nervousness, headache)Fever/hyperpyrexia.

To calculate the reporting rate, the number of reports was used as the numerator and the number of subjects recruited in the study was used as the denominator; the proportion was subsequently multiplied by 100 and the confidence interval at 95% (95%CI) were reported.

For all tests, a two-sided *p*-value < 0.05 was considered statistically significant.

The research conducted for this study was carried out in accordance with the Helsinki Declaration.

## 3. Results

A total of 1481 HCWs were vaccinated during the 2019/20 influenza season; of these, 775 (52.3%) joined on a voluntary basis the active surveillance program. Seven days after the vaccination, 741 of 775 reported the recorded adverse reactions (response rate: 95.6%), and so, they were enrolled in the study; 412 of 741 (55.6%) were female, and the average age was 41.3 ± 14.1, of which 380 (51.3%) were under 40 years old. Of these, 482 of 741 (65.0%) reported having had the flu vaccination also in the 2018/2019 season, and 425 of 741 (57.4%) reported having had the flu vaccination at least once in a previous season. A flu-like syndrome during the 2018/2019 season was reported by 209 of 741 patients (28.2%).

Considering that more than one sign/symptom may also be reported in individual reports, 430 AEFIs (reporting rate: 58.0 (95%CI: 54.4–61.6) × 100 enrolled) were recorded. Of these, 429 of 430 (99.8%; reporting rate: 57.8 (95%CI: 54.2–61.5) × 100 enrolled) were classified as not serious and 1 (0.2%; reporting rate: 0.13 (0.03–0.75) × 100 enrolled).

The recorded AEFIs are described in Table 1; local reactions were the adverse events reported most frequently.

Focusing on local reactions, pain on injection site was the symptom reported most frequently (Table 2).

The recorded AEFIs, stratified by sex and age, are described in Table 3 and Table 4.

The time of insurgence of the reported AEFIs is described in Figure 1; the time interval between 0 and 12 h after vaccination is the one in which the onset of the highest number of adverse reactions was reported (*n* = 285/430; 66.3%) and the time interval between 24 and 48 h after vaccination is the one in which the resolution of the highest number of AEFIs was recorded (*n* = 155/430; 36.0%).

The adverse event classified as serious led to hospitalization of the subject; the onset of the event occurred on the same day as the vaccination, while the patient’s access to the hospital took place two days later. The serious AEFI was characterized by asthenia, myalgia, somnolence (frequently reported after administration of Flucelvax), gastrointestinal disorders (nausea and diarrhea), and symptoms of respiratory tract infection with cough. Causality assessment excluded the causal link between the serious adverse event and the administration of the vaccine; indeed, the diagnostic tests performed during hospitalization showed the presence of a viral infection, probably pre-existing to the administration of the vaccine.

All the AEFIs, including the serious one, were resolved without sequelae.

## 4. Discussion

Our study showed the absolute safety of Flucelvax; almost all recorded AEFIs were not serious (99.8%), except for one serious event (reporting rate: 0.13 × 100 enrolled), which was not associated to the vaccine as excluded by the causality assessment. These data are extremely encouraging, significantly confirming the safety profile of the vaccine, as suggested by other similar studies [18,19].

Local reactions were recorded in almost 90% of enrolled subjects; allergic reactions were very rare (0.2%) and resolved in loco without sequelae. Most of the reactions have been reported in the first 12 h after vaccination and resolved after 24–48 h. No events out of the expected were recorded.

The AEFIs reporting rate in our sample was of 58.0 × 100 enrolled, which was similar to the one reported by Newes Adeyi G et al. [20] (62.0 × 100 enrolled) who reported the results of an active surveillance after influenza vaccine developed by telephonically counselling for 14 consecutive days following immunization. Indeed, it is expected that active surveillance data highlight frequencies of adverse events higher than passive surveillance systems, which are frequently characterized by lower sensitivity and a large component of under-notification [21,22].

Regarding sex, females reported more AEFIs than males, especially local reaction, general malaise, and fever. Even younger persons showed a higher reporting of AEFIs, especially local reactions and gastrointestinal disorders. Sexual and age dimorphism in the human immune system are known [23], confirming the reliability of our results.

One of the strengths of our study is the value of the response rate (94%), which is higher than the one reported by similar studies (ranging from 63% to 90%) [24,25]. Moreover, the homogeneous sample allowed studying the safety of the vaccine on a specific subgroup population. Furthermore, our study confirmed that the implementation of active surveillance programs allows achieving the objectives of defining and monitoring the safety profile of the vaccine; this aspect is particularly relevant in the case of the flu vaccine, for which pre-registration trials have a shorter duration than other vaccines and the general goals of improving system performance. Finally, to our knowledge, no other study specifically investigated the safety of Flucelvax Tetra. The major limit is the sample size that does not allow individuating eventually rare AEFIs. Future studies should provide a larger sample size and a longer follow-up.

In 2014, the Italian Medicines Agency suspended, as a precautionary measure, the use of two batches of adjuvanted influenza vaccine (Fluad, Novartis, Basilea, Switzerland) after three post-vaccination deaths; it trigged a media response that resulted in an increase in the reporting of suspected serious AEFIs after administration of the influenza vaccine [26]. As a consequence, the results of the vaccination campaign were disastrous. Active post-marketing surveillance studies can help to increase the confidence of the population in the safety and efficacy of the influenza vaccine, as the others.

In conclusion, Flucelvax Tetra showed a profile of high safety. Due to their characteristics of greater sensitivity than passive surveillance, active surveillance programs can be useful in defining the safety profiles of a given vaccine/drug in certain population subgroups. The increase in vaccination compliance by the various risk groups also involves defining and knowing how to communicate the adverse effects after vaccination and the probability that they arise. Furthermore, it must be considered that concerning about vaccine safety is a determinant of vaccination hesitancy among health personnel [27], and so, an ad hoc surveillance program should be improved to demonstrate with data the safety and benefits of immunization.

## Figures and Tables

**Figure 1 vaccines-09-00456-f001:**
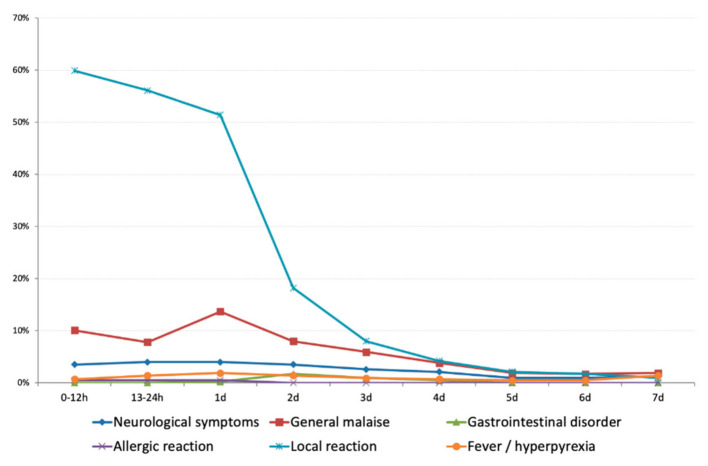
Distribution (%) of recorded AEFIs, per time of insurgence.

**Table 1 vaccines-09-00456-t001:** Recorded AEFIs and reporting rates.

AEFI	*n*	%	Reporting Rate (95%CI) × 100 Enrolled
Local reaction	378	87.8	51.0 (47.3–54.7)
General malaise	114	26.5	15.4 (12.9–18.2)
Neurological symptoms	36	8.4	4.9 (3.4–6.7)
Fever/hyperpyrexia	21	4.8	2.8 (1.8–4.3)
Gastrointestinal disorder	9	2.1	1.2 (0.6–2.3)
Allergic reaction	2	0.4	0.27 (0.03–0.97)
At least one AEFI	430	100.0	58.0 (54.4–61.6)

**Table 2 vaccines-09-00456-t002:** Focus on local reactions and reporting rates.

AEFI (Only Local Reaction)	*n*	%	Reporting Rate (95%CI) × 100 Enrolled
Pain at injection site	357	94.4	48.2 (44.5–51.8)
Induration of injection site	81	21.4	10.9 (8.8–13.4)
Swelling	28	7.4	5.1 (2.5–5.4)
Redness	21	5.5	2.8 (1.8–4.3)

**Table 3 vaccines-09-00456-t003:** Recorded AEFIs and reporting rates, per sex.

	Females (*n* = 412)		Males (*n* = 369)		*p*-Value
AEFI	*n*	Reporting Rate (95%CI) × 100 Enrolled	*n*	Reporting Rate (95%CI) × 100 Enrolled	
Local reaction	240	58.3 (53.3–63.1)	138	41.9 (36.6–47.5)	<0.0001
General malaise	79	19.2 (15.5–23.3)	35	10.6 (7.5–14.5)	0.001
Neurological symptoms	23	5.6 (3.6–8.3)	13	4.0 (2.1–6.7)	0.305
Fever/hyperpyrexia	16	3.9 (2.2–6.2)	5	1.5 (0.5–3.5)	0.054
Gastrointestinal disorder	6	1.5 (0.5–3.1)	3	0.5 (0.2–2.6)	0.501
Allergic reaction	1	0.2 (0.0–1.3)	1	0.3 (0.0–1.7)	0.873
At least one AEFI	270	65.5 (60.7–70.1)	160	48.6 (43.1–54.2)	<0.0001

**Table 4 vaccines-09-00456-t004:** Recorded AEFIs and reporting rates, per age class.

	<40 Years Old (*n* = 380)		≥40 Years Old (*n* = 361)		*p*-Value
AEFI	*n*	Reporting Rate (95%CI) × 100 Enrolled	*n*	Reporting Rate (95%CI) × 100 Enrolled	
Local reaction	229	60.2 (55.1–65.2)	149	41.3 (36.1–46.5)	<0.0001
General malaise	64	16.8 (13.2–21.0)	50	13.9 (10.5–17.8)	0.259
Neurological symptoms	20	5.3 (3.2–8.0)	16	4.4 (2.6–7.1)	0.599
Fever/hyperpyrexia	15	3.9 (2.2–6.4)	6	1.7 (0.6–3.6)	0.061
Gastrointestinal disorder	1	0.3 (0.0–1.5)	8	2.2 (1.0–4.3)	0.015
Allergic reaction	2	0.5 (0.1–1.9)	0	0.0 (0.0–1.0)	0.168
At least one AEFI	251	66.1 (61.0–70.8)	179	49.6 (44.3–54.9)	<0.0001

## Data Availability

Data available upon request to the corresponding author due to restrictions e.g., privacy or ethical.

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
