# Peer review of "Post-Marketing Active Surveillance of Adverse Reactions Following Influenza Cell-Based Quadrivalent Vaccine: An Italian Prospective Observational Study"

_vaccines, 2021, doi:10.3390/vaccines9050456_

Round 1

Reviewer 1 Report

The manuscript titled “Post-marketing active surveillance of adverse reactions following influenza cell-based quadrivalent vaccine: an Italian prospective observational study” is a perspective observational study evaluating, through a post-marketing active surveillance program developed during 2019/20 influenza season, any Adverse Events Following Immunization (AEFIs) happened in the 7 days after immunization with Flucelvax Tetra, which is a cell-based quadrivalent vaccine. This is an important field of study as active surveillance programs can be useful in defining the safety profiles of a given vaccine/drug in certain population subgroups. The manuscript has been written very well and shows the importance of active surveillance program in defining the safety profiles of a given vaccine.

Minor Comments:

  1. Please include the age group/gender of the HCW that were part of this study.
  2. Please break down the data based on age groups/Gender. This may indicate if the AEFI’s were higher in certain age groups/certain population.
  3. Please increase the font size of Figure 1. In the current form it is very difficulty to read the figure data.

Author Response

A1. Revided

A2. Revised

A3. Revised

Reviewer 2 Report

This is a well-written paper addressing an interesting topic.  My interest is enhanced by the fact that the almost identical study could be applied for the different COVID-19 vaccinations.  I am particularly impressed by the response rate.  

My only negative comment relates to Figure 1.  The scale on the horizontal axis is not consistent.  The time span for the first 2 12 hour points is identical to the subsequent one-day time marks.  This affects the shape of the curves.

a minor point at line 43.  provided not providec.

Author Response

A1. Revised

A2. Revised